# DIALS: Dynamic Layer-Skipping Framework for Diffusion Language Models

## Abstract

Diffusion language models (DLMs) have emerged as promising alternatives to autoregressive models (ARMs) due to their bidirectional attention and parallel decoding. However, their inference cost becomes significantly higher as they scale. Layer skipping addresses this challenge by selectively omitting redundant layers. While these dynamic approaches are effective in ARMs, they cannot be naturally extended to DLMs because their parallel generation paradigm makes fine-grained token-level routing challenging. We propose DIALS, a novel dynamic layer-skipping framework for DLMs. DIALS places a lightweight router before each Transformer layer, aggregating masked token representations to make a unified, sequence-level decision on whether to skip or execute the layer. Evaluated on LLaDA-8B across six benchmarks, DIALS generally achieves a better FLOPs-accuracy trade-off compared to static and random layer-skipping baselines. On PIQA, it reduces inference FLOPs by 14.26% without losing accuracy. Our analysis further shows that initial layers are consistently important. Additionally, by incorporating a scaling term based on the mask ratio into the routing objective, we reveal that inherent layer redundancy emerges as denoising progresses.

## 1 Introduction

Diffusion language models (DLMs) (Austin et al., 2021a; Shi et al., 2024; Nie et al., 2025) have emerged as promising alternatives to autoregressive models (ARMs). While ARMs generate tokens from left to right, DLMs generate text through an iterative denoising process over a masked sequence. This characteristic naturally enables bidirectional attention and parallel decoding, which not only mitigates inherent ARM limitations such as the reversal curse (Berglund et al., 2023), but also offers the potential for significantly faster text generation (Nie et al., 2025). Motivated by these architectural advantages, recent research has successfully scaled DLMs up to 8B parameters. Notably, models such as LLaDA (Nie et al., 2025) and Dream (Ye et al., 2025) demonstrate accuracy comparable to similarly sized ARMs.

Scaling up these models introduces a common challenge across LLMs: high computational cost during inference. Since the dense computations within Transformer layers dominate this burden, we explore the potential of layer skipping in DLMs. This technique exploits inherent layer-depth redundancy by selectively omitting the execution of specific layers, thereby reducing the architectural complexity and overall computational cost. In particular, we focus on dynamic layer skipping. Operating on the principle that tasks or tokens of varying difficulty do not require identical computing resources, this approach adaptively bypasses redundant computations based on the specific input.

While dynamic layer skipping has achieved remarkable success in ARMs (Fan et al., 2024; Raposo et al., 2024; Jiang et al., 2024), its application to DLMs remains largely unexplored, with existing research limited to a static approach (Goel et al., 2026). Directly adapting existing ARM-based dynamic mechanisms to DLMs is non-trivial due to a fundamental architectural difference regarding how intermediate layers process inputs. In ARMs, layer execution and routing decisions apply only to the last generated token during decoding. In contrast, DLMs process a fixed-length sequence of tokens simultaneously at each denoising step. This

parallel paradigm prevents the naive application of token-level routing from ARMs, necessitating a shift to sequence-level decisions.

To bridge this gap, we propose DIALS, **d**iffusion language model **i**nference with **a**daptive **l**ayer **s**kipping. DIALS introduces dynamic layer-skipping to DLMs. Specifically, we place a lightweight router before each Transformer layer. Instead of per-token routing, the router aggregates masked token representations to make a unified, sequence-level skip decision. To validate DIALS, we use LLaDA-Base-8B (Nie et al., 2025) and conduct comprehensive experiments across six benchmark datasets. Our empirical analysis reveals a favorable trade-off between computational cost (FLOPs) and accuracy, while we also observe that selectively skipping certain layers often maintains or even improves accuracy, which indicates that executing all layers is not always optimal. Our main contributions are summarized as follows:

- **Dynamic Layer-Skipping Framework for DLMs:** To the best of our knowledge, DIALS is the first framework to achieve dynamic layer skipping in DLMs by aggregating masked token representations for unified, sequence-level routing. We implement this mechanism via a lightweight router to selectively skip redundant Transformer layers.

- **Better FLOPs-Accuracy Trade-offs:** Through evaluations, we demonstrate that DIALS generally provides a better trade-off between computational cost and accuracy compared to static and random layer-skipping methods. Notably, on specific tasks such as PIQA (Bisk et al., 2020), DIALS reduces FLOPs by up to 14.26% while maintaining the original model's accuracy.

- **In-depth Analysis of Dynamic Routing Behavior:** We provide a detailed analysis of how DIALS operates within the iterative denoising process. Our investigation reveals that layer execution patterns are highly task-dependent and that initial layers generally exhibit higher importance. Furthermore, we demonstrate the effectiveness of our training objective, which enables the router to adapt to the varying difficulty of each denoising step.

## 2 Related Work

### 2.1 Diffusion Language Models

Diffusion models (Sohl-Dickstein et al., 2015; Ho et al., 2020; Song et al., 2020) have achieved remarkable success in continuous data generation, including images and video (Rombach et al., 2022; Peebles & Xie, 2023; Ho et al., 2022). In natural language processing (NLP), masked diffusion models (MDMs) (Austin et al., 2021a; Shi et al., 2024; Ou et al., 2024) have emerged as a promising extension of this paradigm. They formulate text generation as an iterative denoising process over an initially masked token sequence. Recent efforts have successfully scaled MDMs up to 8B parameters. Models such as LLaDA (Nie et al., 2025) and Dream (Ye et al., 2025) demonstrate that DLMs can achieve generation quality comparable to that of ARMs (Grattafiori et al., 2024; Yang et al., 2025) of the same size.

On the other hand, scaling to billions of parameters inevitably introduces significant computational and memory overhead during inference. Prior work has explored various efficient inference strategies. These include accelerating inference via caching strategies (Ma et al., 2025; Liu et al., 2025; Wu et al., 2025; Hu et al., 2025; Zhu et al., 2026) and step reduction techniques (Wu et al., 2025; Wei et al., 2025; Hu et al., 2025; Israel et al., 2025), as well as enhancing generation quality (Lu et al., 2025; Wang et al., 2025). In contrast, we take a fundamentally different approach: we propose DIALS to directly reduce the computational cost of DLMs by exploiting architectural redundancy through dynamic layer skipping.

### 2.2 Layer Skipping

Layer skipping is an effective technique for reducing computational cost in LLMs by identifying and omitting redundant layers, broadly falling into static (Fan et al., 2019; Yang et al., 2024; Anwar et al., 2017) and dynamic approaches (Schuster et al., 2022; Xin et al., 2020; Kim & Cho, 2021). Static layer skipping permanently removes layers based on predefined metrics. While this simplicity effectively reduces computational cost without routing overhead, high skipping ratios lead to severe accuracy degradation. In contrast,

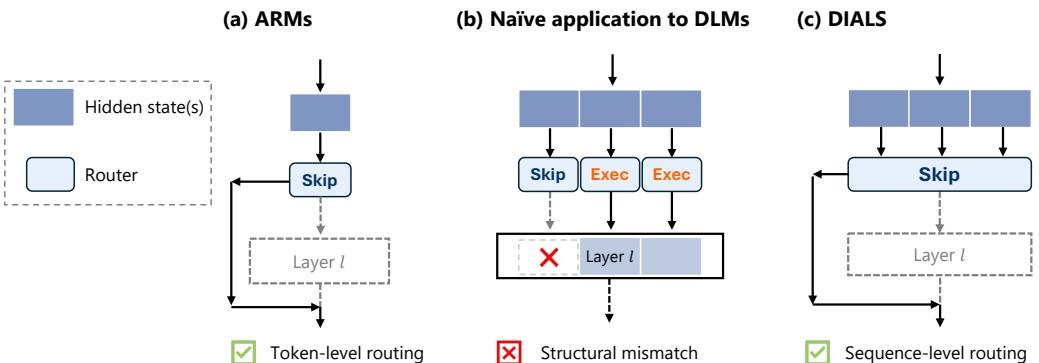

Figure 1: Conceptual comparison of dynamic layer-skipping mechanisms. (a) Standard ARMs support token-level routing naturally due to sequential generation. (b) A naïve application of token-level routing to DLMs results in a structural mismatch, as independent token paths disrupt the dense self-attention mechanism. (c) DIALS introduces sequence-level routing, which aggregates hidden states to make a unified skipping decision for the entire sequence, ensuring structural integrity.

dynamic layer skipping adaptively skips layers based on specific inputs during inference. Early exiting is one such technique (Bolukbasi et al., 2017; Huang et al., 2017; Teerapittayanon et al., 2016), which halts computation entirely at a certain depth for simpler inputs. For NLP, this adaptivity aligns well with the inherent variation in contextual complexity. Consequently, dynamic layer skipping has been widely explored in ARMs (Fan et al., 2024; Raposo et al., 2024; Jiang et al., 2024) , demonstrating remarkable superiority over static approaches (Kim et al., 2024; Gromov et al., 2024) in reducing computational costs. A well-established strategy among these dynamic approaches is employing a lightweight router to make dynamic skipping decisions (Fan et al., 2024; Jiang et al., 2024).

Despite its success in ARMs, dynamic layer skipping remains largely unexplored in DLMs, where research has so far been limited to a static method (Goel et al., 2026). As we detail in Section 3, directly applying ARM-based routing to DLMs is fundamentally hindered by their distinct generation paradigms. This critical gap motivates DIALS, our proposed sequence-level layer-skipping framework that employs a specialized lightweight router.

## 3    Method

We first highlight the fundamental challenge of adapting dynamic layer skipping from ARMs to DLMs. In standard ARMs, token generation proceeds sequentially. Because tokens are appended and processed independently at the last position during decoding, existing dynamic methods (Fan et al., 2024; Raposo et al., 2024; Jiang et al., 2024) can easily apply fine-grained, token-level routing. In contrast, DLMs update the entire token sequence simultaneously at each denoising step. Since their Transformer layers compute dense self-attention over this full sequence, executing a layer on only a subset of tokens creates a severe structural mismatch (Figure 1). Consequently, routing decisions in DLMs are more naturally unified at the sequence level. Guided by this constraint, we design DIALS, a framework that executes or skips a layer for the entire sequence based on the aggregated hidden states of the masked tokens. To formalize this approach, we first review the standard inference process of DLMs in Section 3.1, and then detail our sequence-level layer-skipping mechanism in Section 3.2.

### 3.1    Preliminaries

Diffusion language models (DLMs) formulate text generation as a discrete denoising process. In this framework, "noise" is represented by a special mask token, denoted as [MASK]. Generation proceeds over discrete steps $t = T, T - 1, \ldots, 0$, starting from a fully masked target sequence and progressively refining it into

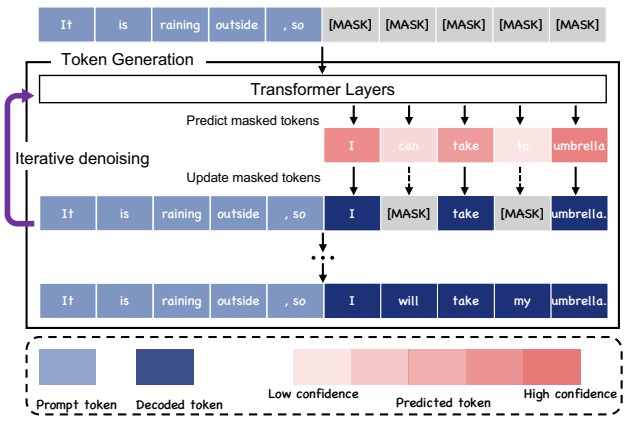

(a) Iterative decoding process of DLMs.

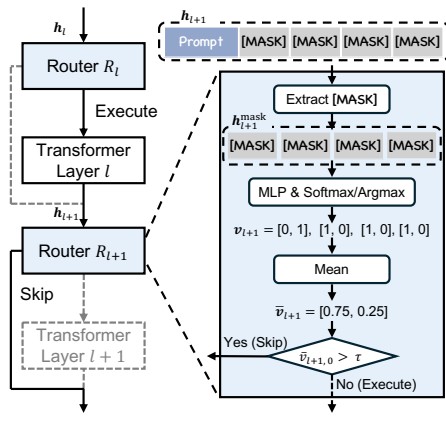

(b) DIALS.

Figure 2: Overview of our proposed framework. (a) Diffusion language models (DLMs) generate text by iteratively updating masked positions through a denoising process. (b) To reduce the computational cost, the router dynamically decides whether to skip the subsequent Transformer layer by evaluating the hidden states of the currently masked tokens.

a clean output. Using LLaDA (Nie et al., 2025) as an example, we formalize the inference process below (Figure 2 (a)).

Let $\mathcal{V}$ be the vocabulary, which includes [MASK]. Given a prompt $\boldsymbol{x} = (x_1, x_2, \ldots, x_n)$ and a user-defined target output length $m$, the initial state $\boldsymbol{x}^{(T)}$ is constructed by appending $m$ mask tokens to the prompt. Thus, the initial sequence of total length $L = n + m$ is defined as:

$$\boldsymbol{x}^{(T)} = (x_1, \ldots, x_n, \underbrace{[\text{MASK}], \ldots, [\text{MASK}]}_{m \text{ tokens}}). \tag{1}$$

During inference, the model progressively denoises this sequence. Let $\boldsymbol{x}^{(t)} \in \mathcal{V}^L$ represent the intermediate sequence at step $t$. At each denoising step $t$, a Transformer-based token predictor $f_\theta$, processes the full sequence and outputs token distribution $\boldsymbol{p}^{(t)} \in \mathbb{R}^{L \times |\mathcal{V}|}$:

$$\boldsymbol{p}^{(t)} = f_\theta(\boldsymbol{x}^{(t)}). \tag{2}$$

For each masked position $i$ (where $x_i^{(t)} = [\text{MASK}]$), the most likely token $\hat{x}_i^{(t)}$ is predicted via greedy decoding:

$$\hat{x}_i^{(t)} = \arg\max(\boldsymbol{p}_i^{(t)}). \tag{3}$$

Finally, a transition function $S$ selectively updates a subset of masked positions in $\boldsymbol{x}^{(t)}$ with the predicted sequence $\hat{\boldsymbol{x}}^{(t)}$ (e.g., based on prediction confidence) to form the next state $\boldsymbol{x}^{(t-1)}$:

$$\boldsymbol{x}^{(t-1)} = S(\hat{\boldsymbol{x}}^{(t)}, \boldsymbol{x}^{(t)}). \tag{4}$$

This iterative process continues until the final generated sequence $\boldsymbol{x}^{(0)}$ contains no mask tokens.

## 3.2  The DIALS Framework

Based on the structural constraints discussed above, we place a lightweight router before each Transformer layer. The overall procedure for dynamic routing during inference is summarized in Algorithm 1. As input to the router, we use only the hidden states of the currently masked tokens. Since these masked tokens are the active targets of generation, this approach naturally aligns with token-level routing in ARMs. By

aggregating these masked token states, the router makes a unified decision on whether to execute or skip the subsequent layer for the entire sequence.

To formalize this routing process, let $\boldsymbol{h}_l \in \mathbb{R}^{L \times d}$ be the sequence of continuous hidden states input to the $l$-th Transformer layer, where $d$ is the hidden dimension. Among these hidden states, let $\boldsymbol{h}_l^{\text{mask}} \in \mathbb{R}^{M_t \times d}$ be the subset of hidden states corresponding to the currently masked tokens, where $M_t$ denotes the number of masked tokens remaining at time step $t$.

As illustrated in Figure 2(b), the router $R_l$ consists of three main components: a two-layer MLP (with RMSNorm and SiLU and a hidden dimension of 512), a token-level decision function, and a sequence-level voting gate. As we analyze later in Section 4.6, this MLP is highly lightweight and introduces negligible computational overhead. Specifically, for each token representation in $\boldsymbol{h}_l^{\text{mask}}$, the MLP outputs two-dimensional logits. To render the routing operation differentiable during training while obtaining strict binary decisions, we apply the Gumbel-Softmax reparameterization trick (Jang et al., 2016) to these logits. During inference, this step simplifies to a straightforward argmax operation. Both approaches convert the logits into a discrete token-level one-hot routing vote, $\boldsymbol{v}_l^{(i)} = [v_{l,0}^{(i)}, v_{l,1}^{(i)}]$ for the $i$-th masked token, where $v_{l,0}^{(i)} = 1$ denotes skip and $v_{l,1}^{(i)} = 1$ denotes execute.

Next, the router aggregates these token-level votes via the mean operation to determine a single, unified sequence-level execution decision. Specifically, we compute the average skip proportion $\bar{v}_{l,0}$ over all $M_t$ masked tokens as follows:

$$\bar{v}_{l,0} = \frac{1}{M_t} \sum_{i=1}^{M_t} v_{l,0}^{(i)}. \tag{5}$$

The final binary gate vector $\boldsymbol{e}_l = [e_{l,0}, e_{l,1}]$ for the entire layer is then determined by comparing this skip proportion with a user-defined threshold hyperparameter $\tau$:

$$e_{l,0} = \mathbb{1}(\bar{v}_{l,0} > \tau), \quad e_{l,1} = 1 - e_{l,0}, \tag{6}$$

where $\mathbb{1}(\cdot)$ is the indicator function. Because this indicator function is non-differentiable, we employ the straight-through estimator (Bengio et al., 2013) during training. This technique allows the model to use the discrete gate values in the forward pass while bypassing the non-differentiable step, routing gradients directly to the continuous proportion $\bar{v}_{l,0}$ during the backward pass. Consequently, the layer execution gate $\boldsymbol{e}_l$ takes a discrete state of either $[1, 0]$ (skip) or $[0, 1]$ (execute) while remaining fully differentiable end-to-end.

Finally, the hidden states output from the $l$-th layer, denoted as $\boldsymbol{h}_{l+1}$, are computed based on the discrete gate values:

$$\boldsymbol{h}_{l+1} = \begin{cases} \boldsymbol{h}_l & \text{if } e_{l,0} = 1, e_{l,1} = 0 \quad (\text{skip}) \\ f_l(\boldsymbol{h}_l) & \text{if } e_{l,0} = 0, e_{l,1} = 1 \quad (\text{execute}), \end{cases} \tag{7}$$

where $f_l(\cdot)$ represents the transformation applied by the $l$-th Transformer layer.

### 3.3 Optimization Objective for Dynamic Routing

During training, we freeze the parameters of the backbone model and update only the newly introduced routers. To ensure stable optimization, the weights of these routers are initialized such that training begins with most of the layers active. To reduce computational cost while maintaining accuracy, we optimize these routers to jointly minimize both the cross-entropy loss and the layer-skipping loss. We define the skip loss as follows:

$$\mathcal{L}_{\text{skip}} = |\alpha - \alpha_{\text{target}}|, \tag{8}$$

where $\alpha_{\text{target}}$ is a predefined target skip rate, and $\alpha$ is the actual layer skip rate. Here, $|\cdot|$ denotes the absolute value. Assuming the model has a total of $N$ Transformer layers, $\alpha$ is computed as the average of the skip proportions across all layers:

$$\alpha = \frac{1}{N} \sum_{l=1}^{N} e_{l,0}. \tag{9}$$

---

**Algorithm 1** DIALS Forward Pass (Inference)

---

**Input:** Initial hidden states $\boldsymbol{h}_1$, Masked token indices $\mathcal{M}$, Threshold $\tau$, Layers $\{f_l\}_{l=1}^N$, Routers $\{R_l\}_{l=1}^N$
**Output:** Final hidden states $\boldsymbol{h}_{N+1}$

1: $M_t \leftarrow |\mathcal{M}|$                                                                         ▷ Number of currently masked tokens
2: **for** $l = 1$ **to** $N$ **do**
3:     $\boldsymbol{h}_l^{\text{mask}} \leftarrow$ Extract representations at indices $\mathcal{M}$ from $\boldsymbol{h}_l$
4:     $\boldsymbol{logits}_l \leftarrow \text{MLP}_l(\boldsymbol{h}_l^{\text{mask}})$                          ▷ Compute 2D logits via MLP of router $R_l$
5:     $\boldsymbol{v}_l^{(i)} \leftarrow \text{OneHot}(\arg\max(\boldsymbol{logits}_l^{(i)}))$ for all $i \in \{1, \ldots, M_t\}$        ▷ Discrete token-level routing vote
6:     $\bar{v}_{l,0} \leftarrow \frac{1}{M_t}\sum_{i=1}^{M_t} v_{l,0}^{(i)}$                          ▷ Compute sequence-level skip proportion
7:     $e_{l,0} \leftarrow \mathbb{1}(\bar{v}_{l,0} > \tau)$                                           ▷ Determine layer skip decision
8:     **if** $e_{l,0} = 1$ **then**
9:         $\boldsymbol{h}_{l+1} \leftarrow \boldsymbol{h}_l$                                              ▷ Bypass layer
10:    **else**
11:        $\boldsymbol{h}_{l+1} \leftarrow f_l(\boldsymbol{h}_l)$                                          ▷ Execute layer
12:    **end if**
13: **end for**
14: **return** $\boldsymbol{h}_{N+1}$

---

During training, the input prompt remains unmasked, and we apply masking exclusively to the target output sequence. Specifically, we randomly replace a subset of target tokens with the mask token according to a mask ratio $p_{\text{mask}}$ uniformly sampled from $\mathcal{U}(0,1)$. Note that we strictly set the training batch size to 1, meaning a new $p_{\text{mask}}$ is independently drawn at each iteration. This randomly set ratio effectively simulates the varying proportions of masked tokens encountered across different denoising steps during inference, rendering the sampling of a denoising step $t$ unnecessary during training. The masked sequence is then fed into the model, where each Transformer layer is dynamically executed or bypassed based on the routers' decisions.

Following the fine-tuning guidelines of LLaDA (Nie et al., 2025), the cross-entropy loss $\mathcal{L}_{\text{CE}}$ for the masked tokens is scaled by the inverse of the mask ratio, $1/p_{\text{mask}}$. Similarly, we introduce an extension by applying the exact same scaling factor to our layer-skipping loss $\mathcal{L}_{\text{skip}}$. The overall objective function $\mathcal{L}$ is thus formulated as:

$$\mathcal{L} = \frac{1}{p_{\text{mask}}}\left(\mathcal{L}_{\text{CE}} + \lambda\mathcal{L}_{\text{skip}}\right), \tag{10}$$

where $\lambda$ is a hyperparameter that controls the trade-off between computational cost and accuracy preservation. This shared scaling mechanism for the skip loss is motivated by the intuition that simpler inputs require fewer layers to process. Smaller sampled values of $p_{\text{mask}}$ correspond to the later stages of denoising, where the generative task inherently becomes easier. Consequently, the amplified $1/p_{\text{mask}}$ term imposes a stronger penalty on computation, encouraging the model to skip more layers during these easier stages.

## 4 Experiments

### 4.1 Setup

We evaluate DIALS on LLaDA-Base-8B (Nie et al., 2025), which consists of 32 Transformer layers. For evaluation, we use six diverse datasets: MMLU (Hendrycks et al., 2020), ARC-C (Clark et al., 2018), Hellaswag (Zellers et al., 2019) and PIQA (Bisk et al., 2020) for general knowledge tasks, GSM8K (Cobbe et al., 2021) for mathematical reasoning, and MBPP (Austin et al., 2021b) for code generation. We train a separate router for each dataset.

During the training of the routers, we set the maximum context length to 1024 and the hidden dimension of the two linear layers to 512. The routers are trained for 10 epochs on PIQA, Hellaswag, and GSM8K, for 20 epochs on MBPP, for 30 epochs on MMLU, and for 40 epochs on ARC-C. To investigate the trade-off between computational cost and accuracy, we fix the target rate $\alpha_{\text{target}}$ to 0.5 in Eq. (8) and experiment with multiple

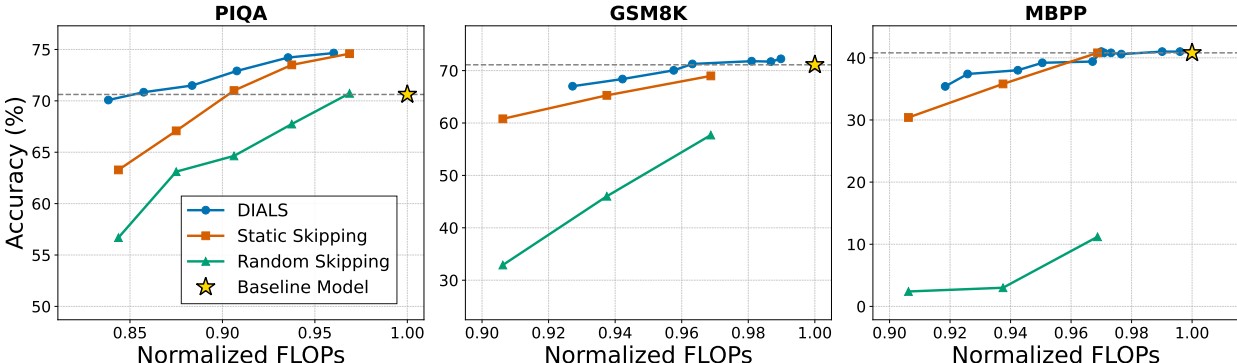

Figure 3: Accuracy vs. normalized FLOPs across three representative datasets (general, math, and code). DIALS (blue) is compared with static (orange) and random (green) layer skipping. The gold star and gray dashed line indicate the original baseline model.

values of the hyperparameter $\lambda$ for each dataset. Additionally, the routing threshold $\tau$ is consistently set to 0.1 across all experiments. For the decoding process, the transition function $S$ in Eq. (4) is implemented using top-1 sampling. Specifically, we set the number of generation steps equal to the generation length, meaning that exactly one token is sampled per denoising step. Finally, regarding the Classifier-Free Guidance (CFG) scale during inference, we adopt a scale of 0.0 (i.e., no guidance) for general knowledge tasks to eliminate potential biases and ensure a fair evaluation of skipped layers. Further details regarding the hyperparameters and experimental settings are provided in Appendix Section A.1.

## 4.2 Efficiency-Accuracy Trade-off

We first evaluate the efficiency-accuracy trade-off of DIALS. Computational cost is measured in FLOPs, reported as a ratio relative to the standard baseline model (where executing all layers equals 1.0). By varying the hyperparameter $\lambda$, we observe a clear trade-off between FLOPs and accuracy. We compare DIALS against two alternative strategies: (1) Static Skipping, which utilizes an algorithm from a recent study (Goel et al., 2026) to statically skip the top-$k$ layers identified as least critical based on the cosine similarity of layer outputs, and (2) Random Skipping, which randomly drops $k$ layers at each denoising step. For these baselines, the trade-off is evaluated by progressively increasing the number of skipped layers, $k$.

Figure 3 illustrates the full trade-off curves across three representative datasets, demonstrating that DIALS generally outperforms both static and random skipping strategies.

While Figure 3 captures the broader trade-off, Table 1 focuses on minimal-degradation configurations across all six datasets to highlight the maximum computational reduction achievable with little to no performance drop. For datasets where DIALS successfully maintains or improves upon the baseline accuracy (i.e., GSM8K, PIQA, ARC-C, and MBPP), we report the configuration with the lowest FLOPs. For static skipping on these datasets, we similarly report the lowest FLOPs that preserves baseline performance, or the $k = 1$ setting if the baseline cannot be maintained. On the remaining two datasets (MMLU and HellaSwag), where any layer skipping consistently degrades performance, we report the DIALS configuration that yields accuracy comparable to the static $k = 1$ setting.

When comparing DIALS to the static layer skipping baseline, we observe that DIALS achieves a superior efficiency-accuracy trade-off on the majority of the datasets (GSM8K, PIQA, MMLU, and MBPP). However, we also note that on ARC-C and HellaSwag, static layer skipping yields slightly better results. This indicates that for certain tasks, a static policy—which consistently preserves specific critical layers—can be robust.

Nevertheless, the overall results demonstrate the high potential of dynamic routing. Interestingly, in four out of the six datasets (GSM8K, PIQA, ARC-C, and MBPP), DIALS successfully maintains or even outperforms the original baseline accuracy while reducing computational cost. Notably, on the PIQA dataset, DIALS

Table 1: Overall FLOPs and accuracy comparison across different tasks.

| Task | Method | Normalized FLOPs ↓ | | Accuracy ↑ | |
|---|---|---|---|---|---|
| | | **Mathematics** | | | |
| GSM8K | Base Model | 1.0000 | | 71.11 | |
| | + Static Skipping | 0.9688 | −3.12% | 68.99 | −2.12 |
| | + DIALS | 0.9632 | −3.68% | 71.27 | +0.16 |
| | | **General Tasks** | | | |
| PIQA | Base Model | 1.0000 | | 70.62 | |
| | + Static Skipping | 0.9063 | −9.37% | 71.00 | +0.38 |
| | + DIALS | 0.8574 | −14.26% | 70.84 | +0.22 |
| MMLU | Base Model | 1.0000 | | 65.85 | |
| | + Static Skipping | 0.9688 | −3.12% | 61.10 | −4.75 |
| | + DIALS | 0.9265 | −7.35% | 61.40 | −4.45 |
| ARC-C | Base Model | 1.0000 | | 44.97 | |
| | + Static Skipping | 0.9375 | −6.25% | 46.16 | +1.19 |
| | + DIALS | 0.9425 | −5.75% | 44.97 | +0.00 |
| HellaSwag | Base Model | 1.0000 | | 70.68 | |
| | + Static Skipping | 0.9688 | −3.12% | 70.38 | −0.30 |
| | + DIALS | 0.9799 | −2.01% | 70.40 | −0.28 |
| | | **Code** | | | |
| MBPP | Base Model | 1.0000 | | 40.80 | |
| | + Static Skipping | 0.9688 | −3.12% | 40.80 | +0.00 |
| | + DIALS | 0.9700 | −3.00% | 41.00 | +0.20 |

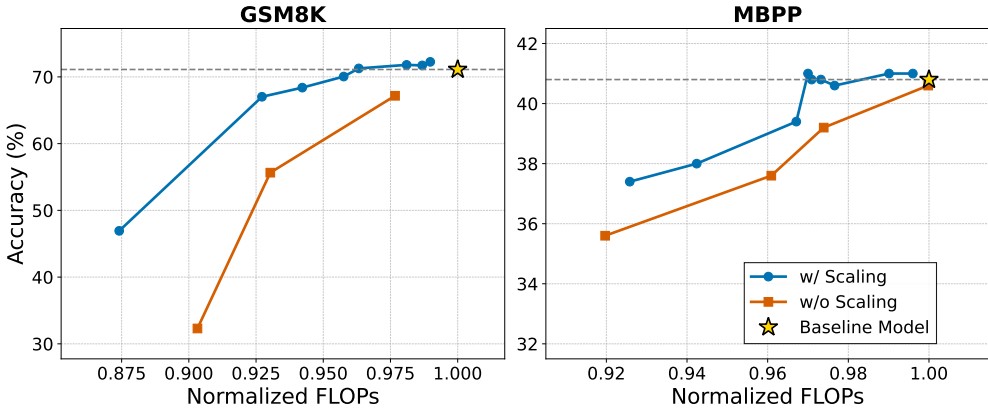

Figure 4: Impact of dynamic scaling on accuracy vs. normalized FLOPs (GSM8K, MBPP). Scaling the skip loss by $1/p_{\mathrm{mask}}$ (blue) maintains higher accuracy by preventing excessive skipping, outperforming the unscaled approach (orange). The gold star and dashed line mark the baseline model.

achieves a 14.26% reduction in FLOPs without any degradation in accuracy. These findings confirm the inherent layer redundancy in DLMs and demonstrate the merit of dynamic layer skipping.

### 4.3 Effect of Dynamic Scaling in the Training Objective

In Section 3.3, we propose a dynamic scaling strategy in our objective function, where the skip loss $\mathcal{L}_{\mathrm{skip}}$ is divided by the current mask ratio $p_{\mathrm{mask}}$. This encourages the routers to skip aggressively when the number of currently masked tokens is small, and to be more conservative when many tokens are still masked.

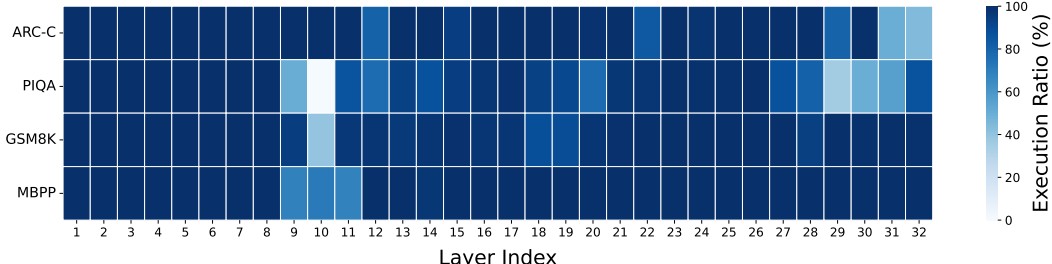

Figure 5: Heatmap of average layer execution ratios across four different datasets. For each task, we visualize the skipping pattern under the configuration that achieves the maximum computational reduction while maintaining baseline performance. Darker blue indicates a higher execution ratio (more computation), while lighter blue indicates a higher skipping ratio (more savings).

To investigate the effect of this scaling strategy, we compare the efficiency-accuracy trade-off curves with and without the strategy. Here, "without scaling" means that the skip loss $\mathcal{L}_{\text{skip}}$ is applied without the $1/p_{\text{mask}}$ factor, meaning its penalty remains independent of the mask ratio.

Figure 4 shows the resulting trade-off curves on GSM8K and MBPP. As shown in the figure, scaling by the inverse mask ratio yields a better efficiency–accuracy trade-off than the unscaled variant on both datasets. These results imply that the early stages of generation might be computationally more demanding, requiring the execution of a large number of layers to process the heavily masked input. Conversely, as the context builds up and fewer tokens remain masked, the prediction task may become easier, which could allow the model to achieve sufficient performance by executing fewer layers.

### 4.4 Analysis of Layer Execution Patterns

To investigate how DIALS operates across different domains, we analyze the execution ratios of each Transformer layer. We select the configurations that achieve the maximum computational reduction (lowest FLOPs) without compromising the baseline accuracy for each of the four datasets.

Figure 5 presents the heatmap of the average execution ratios for all 32 layers under these configurations. The results reveal clear domain-specific execution patterns.

First, the initial eight layers (layers 1–8) are consistently executed at exactly 100% across all evaluated tasks. This suggests that these initial layers are essential for extracting fundamental feature representations, regardless of the task type. Beyond this initial phase, the execution patterns begin to diverge. For instance, in PIQA and GSM8K, the execution ratio drops sharply at layer 10 (to approximately 1% and 40%, respectively), whereas ARC-C maintains 100% execution up to layer 11.

Furthermore, the execution patterns in the subsequent layers reflect the nature of the tasks. For general knowledge and reasoning tasks like PIQA and ARC-C, the execution ratios decrease significantly in the final layers (e.g., dropping to roughly 36–56% in layers 29–31 for PIQA, and 44–50% in layers 31–32 for ARC-C). This indicates that the model can often complete these tasks without relying on the final refinement layers.

In contrast, for tasks requiring multi-step reasoning, such as mathematics (GSM8K) and code generation (MBPP), the router maintains near 100% execution in the latter half of the layers (e.g., layers 16–32). While these tasks allow for some computational reduction in the intermediate layers, they demonstrate a strong reliance on these latter layers. This implies that complex reasoning heavily utilizes the final refinement layers to construct logical sequences and generate accurate outputs.

### 4.5 Analysis of Router Configuration

To investigate whether our baseline router configuration is sufficient for appropriate routing, we analyze how the efficiency-accuracy trade-off changes as the router architecture scales on the ARC-C dataset. In Figure 6(a), we illustrate the trade-off curves for different hidden dimensions (256, 512, and 1024). The results

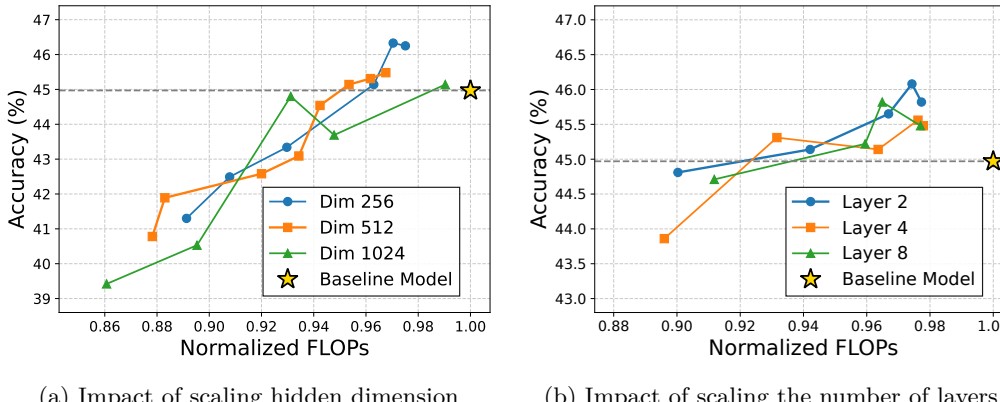

(a) Impact of scaling hidden dimension          (b) Impact of scaling the number of layers

Figure 6: Trade-off between computational cost and accuracy when scaling the router's architecture. (a) Comparing hidden dimensions shows that varying the dimension has little to no impact on the trade-off. (b) Increasing the number of layers from 2 to 4, and up to 8, similarly yields no significant performance changes. Both observations indicate that the baseline capacity is sufficient.

Table 2: Overhead information of the router. The latency and FLOPs metrics are derived from the average performance on the GSM8K dataset.

| Metric | Baseline (LLaDA-8B) | Router Overhead | Relative Increase |
|---|---|---|---|
| Parameters | $\sim$8.0 B | $\sim$0.13 B | + 1.68% |
| Inference Memory | 16.86 GB | + 0.12 GB | + 0.71% |
| FLOPs (per prompt) | $\sim$4,200 TFLOPs | $\sim$120 GFLOPs | + 0.003% |

show that varying the hidden dimension has little impact on the efficiency-accuracy trade-off. Similarly, Figure 6(b) demonstrates that increasing the number of linear layers from 2 to 4, and even further to 8, also does not significantly alter the efficiency-accuracy trade-off. Thus, deeper routers do not improve routing performance. Overall, these observations indicate that increasing the number of router parameters has a negligible effect on performance. Thus, our baseline configuration (2 layers with a hidden dimension of 512) is sufficient for accurate routing.

### 4.6 Router Overhead

We evaluate the overhead of the router relative to the baseline DLM, LLaDA-8B, in Table 2. As shown, the router introduces only a marginal increase in parameter count (under 1.7%). Similarly, the additional peak memory footprint during inference is merely 0.12 GB, which is practically negligible compared to the 16.86 GB required by the baseline DLM.

Furthermore, computational FLOPs and latency vary across individual prompts, as they depend on the sequence length and specific decoding dynamics. However, because the router's computation scales proportionally with the backbone's computation, the relative FLOPs overhead remains consistently low at approximately 0.003%. For instance, on the computationally demanding GSM8K dataset, the router adds an average of roughly 120 GFLOPs to the baseline's $\sim$4,200 TFLOPs per prompt. These results demonstrate that the router overhead is exceptionally modest relative to the backbone cost.

## 5 Limitations and Future Work

We introduce DIALS, a novel dynamic layer-skipping framework for DLMs. While DIALS achieves a generally favorable efficiency-performance trade-off, our current evaluation remains limited to LLaDA-8B, and experiments show that the static baseline is still more robust on certain tasks. Furthermore, the generaliz-

ability of DIALS across a broader range of tasks and other DLMs (e.g., Dream (Ye et al., 2025)) requires further verification. Since DLMs generally share a common training paradigm that simulates inference-time decoding steps by varying mask ratios, our framework can be naturally extended to these models. Additionally, while the current sequence-level routing mechanism based on the aggregated hidden states of masked tokens has proven effective, incorporating additional features could further refine routing decisions. Future research includes exploring more sophisticated routing strategies and analyzing how context length or block size affects skipping rates. Because DIALS directly reduces architectural depth, it remains orthogonal to existing techniques like caching or step reduction. Exploring the combination of these approaches is a promising avenue for future work. Finally, while our empirical analysis suggests that inherent layer redundancy emerges as denoising progresses, uncovering the analytical foundations of these internal dynamics represents an important direction for future research.

## 6  Conclusion

In this paper, we propose DIALS, a novel dynamic layer-skipping framework for reducing the computational cost of inference in diffusion language models (DLMs). To address the structural constraints of DLMs, DIALS employs a sequence-level routing mechanism that aggregates masked token representations to adaptively skip redundant Transformer layers. Through extensive experiments using LLaDA-Base-8B across six diverse datasets, we demonstrate that DIALS generally achieves a superior trade-off between computational cost and accuracy compared to static and random layer-skipping methods. Notably, on the PIQA dataset, DIALS successfully reduces total FLOPs by 14.26% while improving accuracy by 0.22%. Overall, our findings indicate that executing all Transformer layers is not always necessary for DLMs, as dynamic skipping can maintain or even slightly enhance accuracy. Furthermore, our analysis reveals that initial layers are consistently more critical across domains, and suggests that the required number of active layers correlates with the mask ratio—meaning fewer layers are sufficient as the denoising process progresses. We hope that our dynamic routing mechanism and analytical findings will inspire the development of more efficient inference strategies for DLMs.

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

## A  Appendix

### A.1  Extended Experimental Setup and Training Details

We provide further details regarding the experimental settings and hyperparameters used for evaluation. Notably, these configurations are maintained consistently across both the baseline model and our proposed method to ensure a fair and rigorous comparison. Following the evaluation protocol of LLaDA-Base, the general multiple-choice tasks (PIQA, ARC-C, Hellaswag, and MMLU) are evaluated using conditional likelihood estimation, whereas the remaining tasks (GSM8K and MBPP) are evaluated using conditional generation. As summarized in Table 3, the hyperparameters are tailored to these two distinct evaluation formulations.

Table 3: Evaluation settings and hyperparameters for each dataset. "–" indicates that the parameter is not applicable to the evaluation format of the specific task. All settings are identical for both the baseline and our proposed dynamic approach.

| Parameter | PIQA | ARC-C | Hellaswag | MMLU | GSM8K | MBPP |
|---|---|---|---|---|---|---|
| Output Length | – | – | – | – | 256 | 256 |
| Denoising Steps | – | – | – | – | 256 | 256 |
| Block Size | – | – | – | – | 256 | 256 |
| Few-shot | 0 | 0 | 0 | 5 | 5 | 3 |
| Greedy Decoding | False | False | False | False | – | – |
| CFG Scale | 0.0 | 0.0 | 0.0 | 0.0 | – | – |
| MC Num | 128 | 128 | 128 | 1 | – | – |

**Details of the Static Layer Skipping Baseline**  For the static skipping baseline (Goel et al., 2026), we apply their proposed layer-evaluation algorithm to our base model, LLaDA-Base-8B, to identify the least critical Transformer layers. The layers identified by the algorithm are then statically skipped across all denoising steps. Note that the 32 Transformer layers are numbered from 1 to 32. As we progressively increase the number of skipped layers $k$, the specific combinations of skipped layer indices are determined as follows:

| Number of skipped layers ($k$) | Skipped Layer Indices |
|---|---|
| 1 | $\{5\}$ |
| 2 | $\{5, 7\}$ |
| 3 | $\{5, 7, 9\}$ |
| 4 | $\{3, 5, 7, 9\}$ |

**Training Details and Computational Resources**   All models are implemented in PyTorch and run on a server equipped with four NVIDIA H100 GPUs. Specifically, we utilize all 4 GPUs for training, while inference is conducted on a single GPU. During training, the base model parameters are kept strictly frozen to exclusively update the proposed dynamic router. We train using `bfloat16` precision, a maximum sequence length of 1024, and an effective batch size of 64 achieved via gradient accumulation. The router is optimized using AdamW ($\beta = (0.9, 0.95)$, weight decay 0.02, max gradient norm 1.0) with a peak learning rate of $5 \times 10^{-5}$. The learning rate schedule consists of a 2-epoch linear warmup followed by a half-cycle cosine decay. Depending on the dataset, a complete training run on our 4-GPU setup takes from 15 minutes up to roughly 2 hours for Hellaswag, the most computationally intensive task.

Furthermore, to illustrate the learning dynamics of the router, Figure 7 presents the trajectories of the average layer execution ratio evaluated on the validation set during training for the ARC-C dataset under two different scaling factors ($\lambda$). As shown in the figures, the execution ratios smoothly transition and converge to stable values as the training steps progress. Notably, comparing $\lambda = 0.1$ (left) and $\lambda = 4.0$ (right), the trajectory under the higher penalty ($\lambda = 4.0$) converges significantly closer to the target ratio. This clear convergence indicates that the router effectively learns a consistent, reliable layer-skipping policy, and that the scaling factor successfully controls the dynamic balance.

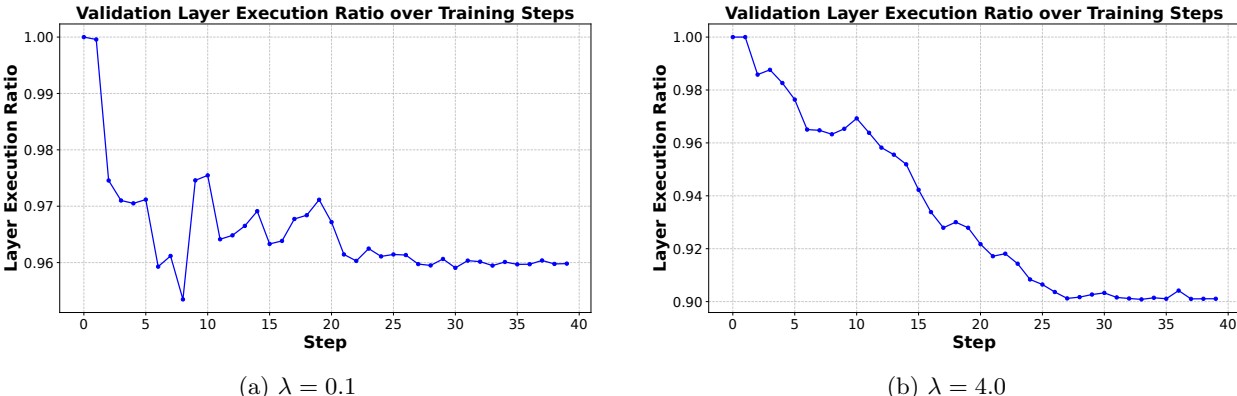

(a) $\lambda = 0.1$          (b) $\lambda = 4.0$

Figure 7: Trajectories of the layer execution ratio on the validation set during the training of the DIALS router on the ARC-C dataset. Both configurations consistently converge to stable values as training progresses. Crucially, the higher penalty factor ($\lambda = 4.0$, right) forces the execution ratio to converge closer to the target compared to the lower penalty ($\lambda = 0.1$, left).

### A.2   Variance Analysis Across Multiple Random Seeds

To further validate the stability and reliability of our findings, we conduct additional experiments using five different random seeds across multiple datasets. In Figure 8, the solid lines represent the mean performance across the five runs, while the shaded regions indicate the standard deviation. As shown, our proposed method consistently maintains its performance advantage regardless of the initial seed.

### A.3   Hyperparameter Stability Analysis of $\alpha_{\text{target}}$ and $\tau$

In our main experiments, we fixed the target skip ratio $\alpha_{\text{target}}$ to 0.5 and the threshold $\tau$ to 0.1, while varying the scaling factor $\lambda$ to control the efficiency-accuracy trade-off. To further investigate the behavior of our framework, we conduct an ablation study on $\alpha_{\text{target}}$ and $\tau$ by fixing $\lambda = 5$ and varying these parameters independently.

Figure 9 illustrates the results. We observe that varying either $\tau$ or $\alpha_{\text{target}}$ does not yield a uniquely optimal configuration. Instead, the models follow the standard efficiency-accuracy trade-off: configurations with lower inference FLOPs consistently show lower accuracy, while those with higher FLOPs achieve higher accuracy. This indicates that although these hyperparameters can also be used to adjust the computational cost, they have little influence on the overall trade-off compared to varying $\lambda$.

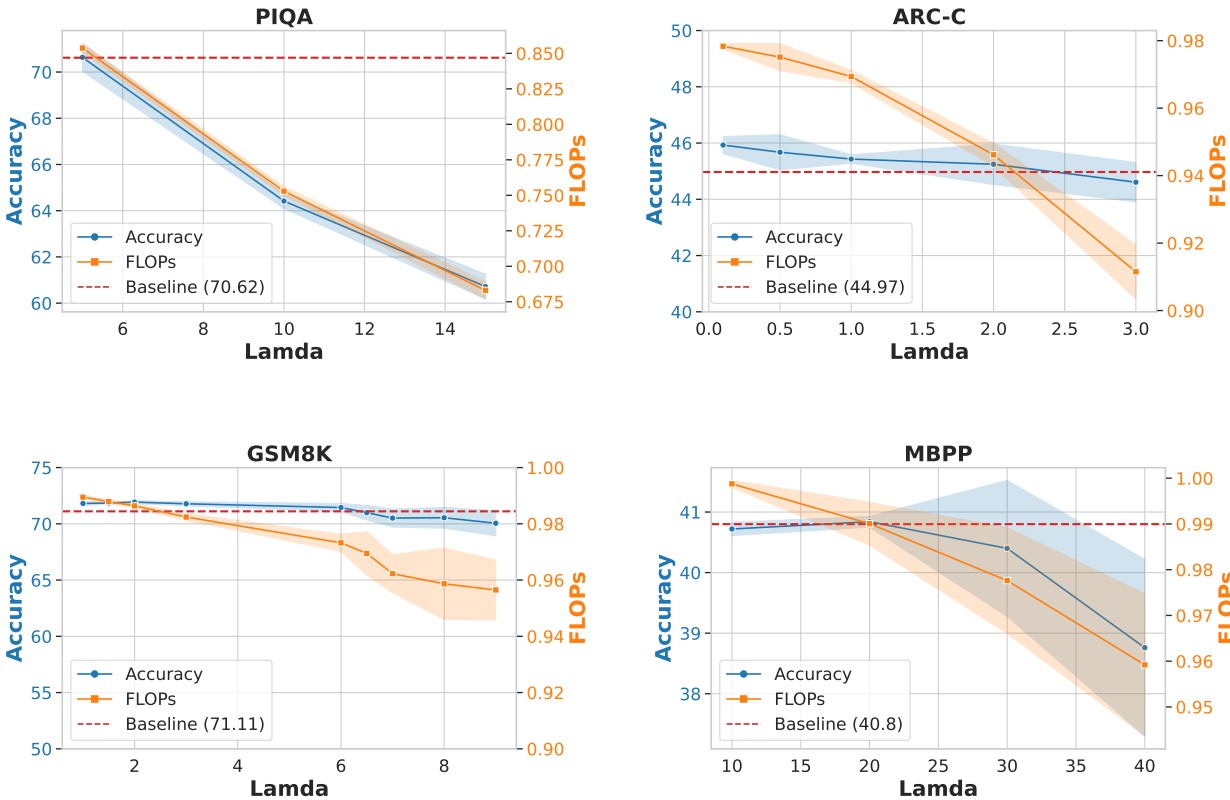

Figure 8: Performance variance across five different random seeds on four datasets. The solid lines indicate the mean, and the shaded regions represent the standard deviation.

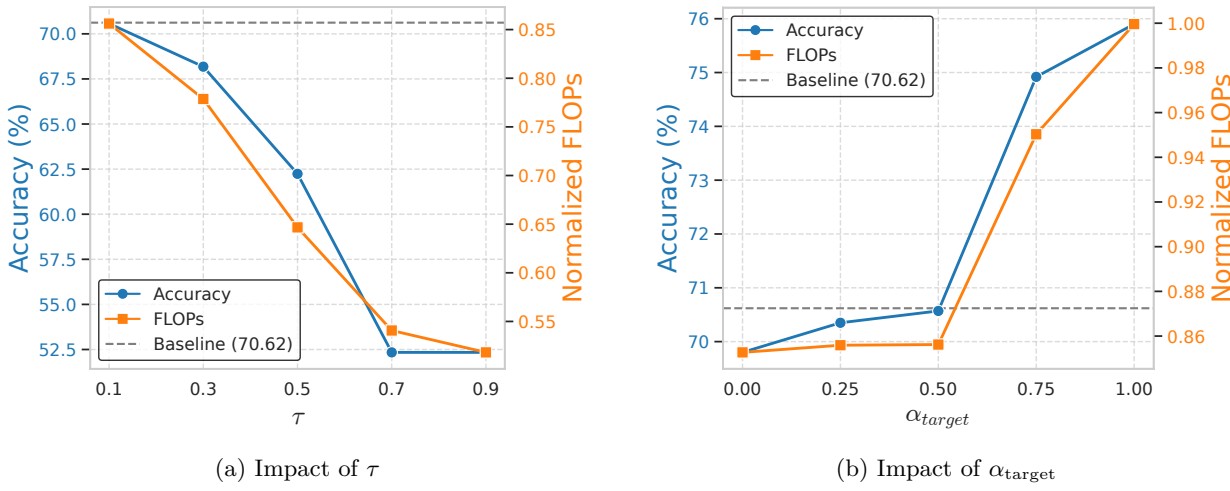

(a) Impact of $\tau$

(b) Impact of $\alpha_{\text{target}}$

Figure 9: Efficiency-accuracy trade-off when independently varying the hyperparameters $\tau$ and $\alpha_{\text{target}}$. Both parameters demonstrate a standard trade-off behavior without exhibiting a uniquely optimal configuration.

Table 4: Cross-task transferability of learned routers. The zero-shot performance is evaluated by transferring routers trained on one specific dataset (columns) to other target tasks (rows). Baseline accuracies (Base Acc) for each evaluated task are provided for reference.

| Evaluated Task | Base Acc ↑ | PIQA Trained | | GSM8K Trained | | ARC-C Trained | |
|---|---|---|---|---|---|---|---|
| | | Acc ↑ | FLOPs ↓ | Acc ↑ | FLOPs ↓ | Acc ↑ | FLOPs ↓ |
| PIQA | 70.62 | 70.57 | 0.8562 | 73.01 | 0.9619 | 75.14 | 0.9852 |
| GSM8K | 71.11 | 71.87 | 0.9922 | 71.19 | 0.9711 | 71.95 | 0.9988 |
| ARC-C | 44.97 | 44.88 | 0.9030 | 42.24 | 0.8977 | 45.14 | 0.9421 |

### A.4 Cross-Task Transferability and Generalization

To investigate the generalization capability of the learned routers and ensure they do not merely overfit to the evaluation datasets, we conducted a cross-task transferability experiment. We evaluated the zero-shot performance of routers trained on specific datasets (PIQA, GSM8K, and ARC-C) across other unseen tasks.

As shown in Table 4, routers trained on general knowledge tasks transfer reasonably well to related domains. For example, the PIQA-trained router evaluated on ARC-C achieved a FLOPs reduction to 90.3% while maintaining an accuracy of 44.88% (closely matching the 44.97% baseline).

Interestingly, when routers trained on common sense reasoning (PIQA or ARC-C) were evaluated on a fundamentally different and rigorous task like GSM8K, the execution ratio naturally converged to nearly 100%. This highlights a robust and safe fallback mechanism: when faced with an out-of-domain task that requires complex multi-step reasoning, the router avoids aggressive skipping, thereby preventing severe accuracy degradation (achieving 71.87% and 71.95% accuracy respectively, successfully maintaining the 71.11% baseline). These findings suggest that the dynamic routing decisions adapt robustly based on the inherent complexity of the input rather than overfitting to a specific skipping ratio.

### A.5 Detailed Latency Profiling and Implementation Efficiency

To demonstrate the real-world efficiency of DIALS and address potential concerns regarding GPU scheduling overhead, we conducted a deep layer-level latency profiling. Throughout our experiments, we observed that a naive dynamic routing implementation in modern frameworks (e.g., PyTorch) incurs significant overhead due to memory slicing (*gather/scatter*) and dynamic control flow synchronization. To mitigate this, we employed a hardware-friendly dense execution strategy. Because the router consists of point-wise linear layers, extracting mask tokens before the MLP is mathematically equivalent to applying the MLP across the dense sequence and extracting the mask token outputs afterward. By adopting the latter, we effectively bypass the memory irregularity before the matrix multiplications, maximizing GPU Tensor Core utilization.

Profiling Results: Under this optimized implementation, taking the GSM8K dataset as an example, the total execution overhead introduced by DIALS per Transformer block is approximately 0.232 ms. Considering that the execution of a pure standard LLaDA-8B block takes roughly 1.453 ms, the dynamic routing introduces merely a ∼16% latency overhead per block. Currently, our framework requires approximately a 16% layer skip rate to break even in actual wall-clock time.

Hardware and Framework Challenges: A detailed breakdown of this 0.232 ms overhead reveals significant future optimization potential, as presented in Table 5. Crucially, the pure neural network computation of the router (linear projections, activations, and softmax) consumes only ∼0.087 ms. This accounts for merely 37.5% of the total DIALS overhead, representing a mere ∼6.0% of the baseline Transformer block's execution time. The remaining majority of the latency (∼62.5%) is not computational but is entirely dominated by framework-level artifacts required to manage dynamic execution within PyTorch. Specifically, supporting dynamic routing across multiple sequences in a batch requires block-level memory slicing (gather and scatter, ∼0.065 ms) to extract only the active batch indices and write back their results. Furthermore, evaluating these dynamic batch sizes requires GPU-to-CPU synchronization (e.g., `torch.nonzero`, ∼0.060 ms). These

Table 5: Detailed latency breakdown of the DIALS overhead per Transformer block (taking GSM8K as an example). The pure neural network computation accounts for only ~37.5% (0.087 ms) of the total overhead, representing a mere ~6.0% of the baseline block's latency. The majority of the overhead is consumed by framework-level artifacts required for dynamic batch routing, specifically block-level memory slicing (gather/scatter) and CPU-GPU synchronization.

| Operation Category | Latency (ms) | Proportion |
|---|---|---|
| **Pure NN Compute** (Linear, Act, Softmax, Logic) | 0.087 | 37.5% |
| **Batch-level Slicing** (Block Gather / Scatter for active batches) | 0.065 | 28.0% |
| **Synchronization** (`torch.nonzero`, GPU-CPU sync) | 0.060 | 25.9% |
| **Token-level Slicing** (Router internal Gather / Scatter) | 0.006 | 2.6% |
| **Other Framework Overhead** | 0.014 | 6.0% |
| **Total DIALS Overhead** | 0.232 | 100.0% |
| *Reference: Pure Baseline Transformer Block* | 1.453 | - |
| *Ratio: Pure NN Compute / Baseline Block* | - | 6.0% |

profiling results highlight that the pure algorithmic complexity of DIALS is remarkably minimal. The current latency-efficiency trade-offs imposed by framework overheads are not fundamental limits, but rather engineering challenges to be overcome. As custom kernels for dynamic execution evolve to fuse batch masking and routing natively, we anticipate the actual wall-clock latency of DIALS will converge closely toward its theoretical FLOPs reduction bounds.

