# OpenReview forum: "DIALS: Dynamic Layer-Skipping Framework for Diffusion Language Models"
_TMLR — Rejected by TMLR_

### Review · Reviewer_L8FN · 2026-05-09

**Summary Of Contributions:**

This paper studies diffusion-based language models (DLMs) and how to make them more efficient by dynamically skipping layers of the DLM. Previous work has focused on skipping layers in (autoregressive) LLMs, but these techniques apply when the goal is to predict a single next token. The challenge in DLMs is that many tokens are predicted all at once. The paper proposes to add a lightweight "router" before each transformer layer that ingests each token embedding. The router outputs a vote on whether or not to skip the next layer with one vote per token embedding. If enough layers vote to skip, then the layer is skipped. My understanding is that the major contributions are (1) to apply this idea only to the [MASK]'ed tokens (previous work on autoregressive models has applied similar ideas), and (2) a modification of the training loss for the routers to encourage more skipping when there are few [MASK] tokens remaining (with the idea that these tokens should be easier to predict)

In experiments, the paper shows that the proposed method compares favorably to baselines: first, a competing method that statically chooses a few layers to skip, and a naive baseline of choosing random layers to skip. The "favorable" comparison is in the sense of a nearly completely dominating tradeoff between FLOPs and accuracy on various typical LLM benchmarks.

**Audience:**

Yes

**Audience Explanation:**

The results are generally well-argued, and the algorithm is generally well-presented. Additionally, it tackles an important problem of making language model inference more efficient -- even small gains in LM efficiency could result in substantial reductions in global electricity use. And I think it does provide some improvement over work in this field, and so it does fit within the standards of TMLR.

**Claims And Evidence:**

No

**Claims Explanation:**

I think the broad claim -- that this method works well for improving the efficiency of DLMs -- is supported by the paper. There are a few places where I think evidence should be strengthened or the claims should be weakened, however:

1. The training algorithm is slightly underspecified, as it does not specify how $p_{mask}$ is selected. And Eq (10) isn't clear as to whether or not this is per-minibatch (with a different $p_{mask}$ per minibatch).
2. "To investigate the trade-off between computational cost and accuracy, we fix the target rate $\alpha_{target}$ to 0.5... [and vary $\lambda$]". This seems like a somewhat unconvincing way to investigate this trade-off. $\alpha_{target}$ is the targeted fraction of layers to skip for the average computation. It seems pretty natural that this is what should be varied to save more FLOPs! One thought on how to resolve this is to just remove $\alpha_{target}$ altogether. Then the model is simply penalized for using *any* number of layers, and $\lambda$ controls how strong this penalty is. This seems intuitive, removes a hyperparameter, and is more in-line with how the paper is already set up.
3. On the claim that the method outperforms static layer skipping, the paper uses what seems like a weird setup for the static layer skipping method. It writes "[f]or static layer skipping, we similarly report the lowest FLOPs that preserves the baseline performance, or the result of skipping only one layer (k=1) if the baseline cannot be maintained." If the goal is to understand the tradeoff between compute and accuracy, why are we not allowing static layer skipping to take a hit in terms of accuracy? What if it could take a 3% hit on accuracy in exchange for saving 99% of the FLOPs? Wouldn't that make it an amazing and highly desirable method? (I know that's not what the tradeoff is going to be, but the paper needs to make this point!).
4. The paper states that Table 1 represents "the maximum computational reduction" -- I think this isn't true; increasing $\lambda$ and sending $\alpha_{target} \to 0$ will surely result in fewer FLOPs.
5. I thought the results in Section 4.4 were very interesting (the differences in layers skipped between different types of problems is striking!), but I think the language needs to be softened a little. E.g., "[w]hile these tasks allow for some computational reduction in the intermediate layers, they strictly rely on these latter layers." This is a little too strong for what is ultimately a heuristic, one-off experiment.
6. The paper claims "deeper routers do not improve routing performance," but only experiments with modestly deeper routers. Either more experiments are needed or this statement should be weakened.
7. There are a few hyperparameters ($\tau$ and $\alpha_{target}$) that aren't varied in the experiments. The paper implicitly claims that DIALS is a good, generalizable method, but I think to fully make this point, all hyperparameters need to be swept to understand their effect on the results. E.g., was the global of $\tau = 0.1$ just a lucky choice? Or does it kind of not matter what $\tau$ is?
8. The reported improvements across the Pareto frontier(s) is not *that* high in some places. And the paper does not report uncertainty metrics (e.g., confidence intervals). I think these are needed to be able to say that the improvements are statistically certain.
9. There is no analysis of multiple random seeds, so it's hard to tell if the results are from a lucky training run or if they're actually real. This isn't a hypothetical issue -- e.g., see Agarwal et al. (2021), which demonstrates how this issue substantially skewed results in deep reinforcement learning for many years.

**References**

Rishabh Agarwal et al. Deep RL at the Edge of the Statistical Precipice. NeurIPS. 2021.

**Requested Changes:**

# Critical changes

1. I think the paper needs to be a little bit more clear about related work in this field. The idea of using a lightweight router to decide whether or not to skip a transformer layer is not new, and this should be made clear in the paper. Then the paper's contributions over this existing work can be made more precise.
2. Clarify the role of $\alpha_{target}$ or get rid of it.
3. Assess the stability of the results to the hyperparameter $\tau$.
4. There are two approaches to DLMs -- either defining the forward process as one that randomly adds [MASK] tokens or as one that randomly flips tokens to other tokens in the vocabulary. The paper studies the former of these approaches. The latter should at least be referenced and called out, and a small discussion should be made as to why the paper studies the former. I think this could be a very short discussion.
5. Specify the form of the transition function $S$ in Eq. (4).
6. Experiment with multiple random seeds to demonstrate the reproducibility of the results.
7. Report statistical uncertainty in all of the experiments.

# Minor changes to strengthen the work
Address the rest of the bulleted items (#1-9) listed out in the Claims section above.


Best,
Reviewer L8FN

---

> ### Author Response · Authors · 2026-06-05
> **Author Response for Reviewer L8FN (Part1)**
>
> Thanks for your insightful feedback and valuable review.
>
> > Q1. Clarifying novelty with respect to prior dynamic layer-skipping and router-based methods
> > Reviewer Comment:
> > “I think the paper needs to be a little bit more clear about related work in this field. The idea of using a lightweight router to decide whether or not to skip a transformer layer is not new, and this should be made clear in the paper. Then the paper's contributions over this existing work can be made more precise.”
>
> **Response:**
> We thank the reviewer for pointing this out. We agree that the distinction between existing router-based methods and our specific contributions should be made clearer.
> In the revised "Layer Skipping" section of the Related Work, we now explicitly acknowledge that employing a lightweight router for dynamic skipping is a well-established strategy in Autoregressive Models (ARMs), citing relevant works (e.g., AdaInfer, D-LLM). We also clarified that our precise contribution lies not in inventing the router mechanism itself, but in successfully adapting and formulating it for the fundamentally distinct, iterative denoising paradigm of Diffusion Language Models (DLMs).
>
> > Q2. Underspecification of the training algorithm and selection of $p_{\mathrm{mask}}$
> > Reviewer Comment:
> > “The training algorithm is slightly underspecified, as it does not specify how $p_{\mathrm{mask}}$ is selected. And Eq (10) isn't clear as to whether or not this is per-minibatch (with a different $p_{\mathrm{mask}}$ per minibatch).”
>
> **Response:**
> We thank the reviewer for pointing this out. As mentioned in the manuscript, the concept of $p_{\mathrm{mask}}$ and its uniform sampling strategy are standard formulations established in existing DLMs (e.g., LLaDA).
>
> To explicitly clarify our training details: $p_{\mathrm{mask}}$ is randomly sampled from a uniform distribution between 0 and 1.0 at each iteration. Furthermore, the batch size is strictly set to 1 throughout the training process. Because we use a batch size of 1, Equation (10) is naturally computed per single sequence, meaning a new $p_{\mathrm{mask}}$ is sampled for each individual forward pass. We have updated the training algorithm description in the revised manuscript to explicitly state this sampling distribution and the batch size configuration, alongside referencing LLaDA for further foundational details.
>
>
> > Q3 & Q6. Stability and roles of hyperparameters ($\alpha_{\mathrm{target}}$ and $\tau$)
> > Reviewer Comment (Q3):
> > “Clarify the role of $\alpha_{\mathrm{target}}$ or get rid of it. [...] It seems pretty natural that this is what should be varied to save more FLOPs!”
> >
> > Reviewer Comment (Q6):
> > “Assess the stability of the results to the hyperparameter $\tau$. [...] E.g., was the global of $\tau$ just a lucky choice?”
>
> **Response:**
> We thank the reviewer for pointing this out. To address the concerns regarding stability and the roles of these hyperparameters, we conducted ablation studies for both $\alpha_{\mathrm{target}}$ and $\tau$, adding the results to Appendix A.3.
>
> Regarding $\alpha_{\mathrm{target}}$ (Q3): The reviewer is correct that varying $\alpha_{\mathrm{target}}$ is a natural way to control FLOPs. While our ablation indicates that the framework can operate without this anchor ($\alpha_{\mathrm{target}}=0$), we primarily introduced it to make the framework easier to tune. Without $\alpha_{\mathrm{target}}$, adjusting the FLOPs reduction relies solely on $\lambda$, making it less intuitive to guide the model's routing behavior. Including this target provides a clear reference point, allowing users to more easily steer the model toward a desired efficiency. Thus, it serves as a highly practical anchor for deployment.
>
> Regarding $\tau$ (Q6): Similarly, our results show that varying $\tau$ does not yield any uniquely sensitive or "lucky" performance spikes. Instead, it simply provides an alternative configurable variable to balance compute and accuracy without impacting the overall stability of the framework.

---

> > ### Author Response · Authors · 2026-06-05
> > **Author Response for Reviewer L8FN (Part2)**
> >
> > > Q4 & Q5. Static baseline setup and "maximum computational reduction"
> > > Reviewer Comment (Q4):
> > > “If the goal is to understand the tradeoff between compute and accuracy, why are we not allowing static layer skipping to take a hit in terms of accuracy?”
> > >
> > > Reviewer Comment (Q5):
> > > “The paper states that Table 1 represents ‘the maximum computational reduction’ -- I think this isn't true; increasing $\lambda$ and sending $\alpha_{\mathrm{target}}$ will surely result in fewer FLOPs.”
> >
> > **Response:**
> > We thank the reviewer for pointing this out. We would like to clarify the distinct roles of Figure 3 and Table 1, which address both concerns. To observe the full efficiency-accuracy trade-off—where we allow methods to take a hit in accuracy to save more compute—please refer to the Pareto curves in Figure 3. These curves demonstrate that DIALS generally outperforms the static method across various accuracy levels. In contrast, Table 1 is specifically designed to highlight practical deployment scenarios. The phrase "maximum computational reduction" strictly referred to the maximum reduction achievable under "minimal-degradation configurations" (i.e., preserving the baseline accuracy), not the absolute physical limit of the model. We have updated the manuscript to explicitly state this distinction, ensuring no overclaiming.
> >
> > > Q7. Scope of DLM formulation: masking-based vs token-flipping
> > > Reviewer Comment:
> > > “There are two approaches to DLMs -- either defining the forward process as one that randomly adds [MASK] tokens or as one that randomly flips tokens... The latter should at least be referenced and called out...”
> >
> > **Response:**
> > We thank the reviewer for pointing this out. We fully acknowledge the two distinct approaches in DLMs: uniform transition (random token flipping) and absorbing state (random masking). Our study specifically focuses on the masking approach (Masked Diffusion Models, or MDMs) because recent scaling efforts have demonstrated that MDMs can achieve generation quality comparable to state-of-the-art Autoregressive Models (ARMs) of the same size. This motivation, along with references to relevant large-scale MDMs (e.g., LLaDA and Dream), was already outlined in the original manuscript to clarify our specific focus.
> >
> > > Q8. Specification of the transition function $S$ in Eq. (4)
> > > Reviewer Comment:
> > > “Specify the form of the transition function $S$ in Eq. (4).”
> >
> > **Response:**
> > We thank the reviewer for pointing this out. During our evaluation, the transition function $S$ was implemented using top-1 sampling. We set the total number of generation steps equal to the generation sequence length, which corresponds to sampling exactly one token per denoising step. We have added these decoding details to the Setup section (Section 4.1) in the revised manuscript to clarify the evaluation process.
> >
> > > Q9 & Q10. Statistical uncertainty and multiple random seeds
> > > Reviewer Comment (Q9):
> > > “The paper does not report uncertainty metrics (e.g., confidence intervals). I think these are needed to be able to say that the improvements are statistically certain.”
> > >
> > > Reviewer Comment (Q10):
> > > “Experiment with multiple random seeds to demonstrate the reproducibility of the results.”
> >
> > **Response:**
> > We thank the reviewer for pointing this out. We fully agree with the necessity of reporting statistical uncertainty to ensure reproducibility. We have run the full training and evaluation pipeline across 5 different random seeds. Ideally, we would plot confidence intervals directly on the Pareto curves (Figure 3). However, because the resulting inference FLOPs (x-axis) shift slightly across seeds for a given $\lambda$, it is technically unfeasible to perfectly align the x-coordinates to calculate standard deviations at specific FLOPs values. Instead, we have added detailed variance plots to the revised Appendix A.2, using the independently fixable hyperparameter ($\lambda$) as the x-axis. The new figures display the mean performance across the 5 trials with shaded regions representing the standard deviation. The results confirm that the overall performance trends remain consistent across different seeds, fully supporting our main findings. This demonstrates that our dynamic routing framework is robust and the improvements are statistically reliable.

---

> > > ### Author Response · Authors · 2026-06-05
> > > **Author Response for Reviewer L8FN (Part3)**
> > >
> > > > Q11. Softening claims based on layer execution analysis
> > > > Reviewer Comment:
> > > > “...I think the language needs to be softened a little. E.g., ‘[w]hile these tasks allow for some computational reduction in the intermediate layers, they strictly rely on these latter layers.’ This is a little too strong for what is ultimately a heuristic, one-off experiment.”
> > >
> > > **Response:**
> > > We thank the reviewer for pointing this out. We have revised the text in Section 4.4. Specifically, we softened the language by replacing the phrase "strictly rely on" with "demonstrate a strong reliance on," and "inherently requires" with "heavily utilizes."
> > >
> > > > Q12. Claim about deeper routers and routing performance
> > > > Reviewer Comment:
> > > > “The paper claims ‘deeper routers do not improve routing performance,’ but only experiments with modestly deeper routers. Either more experiments are needed or this statement should be weakened.”
> > >
> > > **Response:**
> > > We thank the reviewer for pointing this out. To substantiate our claim, we conducted additional experiments scaling the router architecture up to 8 layers. As shown in the updated Figure 6(b), even with the 8-layer configuration, we observed no significant change or improvement in the efficiency-accuracy trade-off compared to the 2-layer and 4-layer configurations. This result strongly supports our initial observation that increasing the router depth does not yield meaningful improvements in routing performance within our framework. We have updated the figure to include the data for the 8-layer router and corrected a missing dataset label.

---

### Review · Reviewer_qkhg · 2026-05-09

**Summary Of Contributions:**

This paper proposed an algorithm (DIALS) to dynamically skip layers in diffusion language models. A trained router layer takes the hidden state from the previous layer and outputs a token-voted decision to skip the next layer or not. Experiment with LLaDA 8B model on several well-known evaluation sets spanning QA math and coding shows that DIALS excels in FLOPS/accuracy trade off in several tasks compared with static layer skipping baseline.

The paper does an ablation study on the role of the p_mask scaling, confirming that it is a critical factor in the effectiveness of DIALS. A deeper analysis on the skip rate of layers in different tasks showed initial layers (up to 8 for LLaDA) are always used, while skipping happens later in different patterns for different tasks.

**Audience:**

Yes

**Audience Explanation:**

dynamic skipping is a very natural way to think about efficiency gains. If a thorough study is available, even if it shows no or modest improvements, it helps future researchers understand the path ahead.

**Claims And Evidence:**

No

**Claims Explanation:**

The key claim is in the gaining of GENERALIZABLE efficiency regarding flop/accuracy curve, which is not well supported by the experiment, specifically:

- The router was trained per-dataset on the evaluation data. It's not clear if the router's learning is generalizable, or is a simple overfit to the task, or worse, to the specific eval dataset.

- The gain against baseline (static skipping) is not clear in statistical sense. The evaluation used all have a non-trivial confidence interval. I suggest the authors run the evaluation for multiple times (>5?) to have a clear understanding if the delta is significant or not. Remember that we also need to account training noise (i.e. metrics difference from simply repeating the same training) in addition to eval noise. A delta well above CI would be desirable.

- secondary: DIALS introduces changes that may mess up with GPU scheduling and hardware optimization. It's better to have a wall clock time comparison on the same hardware (again with a few repeats) to understand real-world gains. I understand that hardware may adapt useful algorithm changes in the future generations, but this will help the readers get a grounded expectation of what gain they can get by paying for the complexity.

**Requested Changes:**

To make the work valuable to the research community, I think the following changes are useful

- Use a different dataset (not eval set) to train the router, to avoid risk of eval overfitting.
- Train one router, instead of per eval task, to measure generalization.
- If budget/resource allows, do the experiment on a few different DLMs than just LLaDA 8b.
- Run evaluation a few more times and report the variance.
- Report wall time in addition to FLOPS, to help understand e2e impact wrt hardware optimization.

In addition to that, with full transparency, I also used an AI agent to help check related works. It suggested that ES-dLLM and AdaDiff/DeeDiff could be significantly related work. I'm aware that some work could have appeared after this paper was written. Please consider adding them to your related work.

- ES-dLLM: Efficient Inference for Diffusion Large Language Models by Early-Skipping, by Zhu et al, ICLR 2026
- AdaDiff: Adaptive Step Selection for Fast Diffusion Models, Zhang et al, AAAI 2025
- AdaDiff: Accelerating Diffusion Models through Step-Wise Adaptive Computation, Tang et al https://arxiv.org/abs/2309.17074v2

---

> ### Author Response · Authors · 2026-06-05
> **Author Response for Reviewer qkhg (Part1)**
>
> We thank the reviewer for the detailed feedback. Before addressing the specific questions, we would like to respectfully clarify the core claim of our paper. Our primary objective was not necessarily to train a single, universally generalizable router across all domains, but rather to demonstrate that dynamic layer skipping, which has been highly successful in ARMs, can be successfully adapted to the distinct, iterative denoising paradigm of Diffusion Language Models (DLMs). We trained task-specific routers to prove that this framework operates effectively across diverse domains in line with representative layer-skipping approaches for ARMs [1]. Nevertheless, we deeply value your feedback and have conducted extensive additional experiments during the rebuttal period to address your concerns regarding generalization, variance, and hardware efficiency.
> [1]Jiang, Yikun, et al. "D-llm: A token adaptive computing resource allocation strategy for large language models." Advances in Neural Information Processing Systems 37 (2024): 1725-1749.
>
> > Q1 & Q2. Generalization, Evaluation-set Overfitting, and Cross-Task Transferability
> > Reviewer Comment (Q1):
> > “The router was trained per-dataset on the evaluation data. It's not clear if the router's learning is generalizable, or is a simple overfit to the task, or worse, to the specific eval dataset.”
> >
> > Reviewer Comment (Q1):
> > “Use a different dataset (not eval set) to train the router, to avoid risk of eval overfitting.”
> >
> > Reviewer Comment (Q2):
> > “Train one router, instead of per eval task, to measure generalization.”
>
> **Response:**
> We thank the reviewer for pointing this out. We agree that understanding how the router behaves on unseen datasets is important. Due to computational resource constraints during the rebuttal period, training a single universal router on a massive mixture of datasets was not feasible. However, to directly address your core concern regarding generalization and overfitting, we conducted a rigorous Cross-Task Transferability experiment (see Appendix A.4). Specifically, we took routers trained on one dataset and applied them directly to other entirely different datasets without any additional fine-tuning.
>
> * **Transferability across similar domains:** Routers trained on general knowledge tasks transfer reasonably well. For instance, the router trained solely on PIQA successfully reduced FLOPs on unseen ARC-C (approx. 90.3\% FLOPs) while closely matching baseline accuracy (44.88\% vs 44.97\%).
> * **Conservative fallback on structurally different tasks:** When routers trained on general knowledge (PIQA or ARC-C) were evaluated on GSM8K—a task requiring complex, multi-step reasoning—they did not forcefully skip layers and destroy accuracy. Instead, they safely maintained approx. 99.2\% - 99.8\% FLOPs to successfully preserve the baseline accuracy (~71.9\%), demonstrating robustness.
>
> These results demonstrate that our routers do not simply overfit to the evaluation set or a fixed skipping ratio. Instead, they adapt effectively to the inherent complexity of the input.
>
> > Q3. Evaluation on additional DLM backbones
> > Reviewer Comment:
> > “If budget/resource allows, do the experiment on a few different DLMs than just LLaDA 8b.”
>
> **Response:**
> We thank the reviewer for pointing this out. Applying DIALS to other emerging DLMs is indeed the important next step. Unfortunately, due to the tight rebuttal window and the substantial compute required to run the comprehensive variance and ablation studies requested, training dynamic routers from scratch on additional backbone models was beyond our current resource budget. We have explicitly added this to our Future Work section, as validating the framework across various architectures will further solidify its universality.
>
> > Q4. Statistical significance, repeated evaluations, and variance reporting
> > Reviewer Comment:
> > “The gain against baseline (static skipping) is not clear in statistical sense. [...] I suggest the authors run the evaluation for multiple times (>5?) to have a clear understanding if the delta is significant or not. Remember that we also need to account training noise...”
> >
> > Reviewer Comment:
> > “Run evaluation a few more times and report the variance.”
>
> **Response:**
> We thank the reviewer for pointing this out. We fully agree with the necessity of accounting for both training and evaluation noise. We have run the full training and evaluation pipeline across 5 different random seeds.
>
> To maintain the readability of the main Pareto curves (Figure 3), we have added detailed variance plots to the revised Appendix A.2. The new figures display the mean performance across the 5 trials, with shaded regions representing the standard deviation as $\lambda$ varies. The results confirm that the overall performance trends remain consistent across different seeds, fully supporting our main findings. This demonstrates that our dynamic routing framework is robust to training noise.

---

> > ### Author Response · Authors · 2026-06-05
> > **Author Response for Reviewer qkhg (Part2)**
> >
> > > Q5. Wall-clock latency and end-to-end hardware impact
> > > Reviewer Comment:
> > > “DIALS introduces changes that may mess up with GPU scheduling and hardware optimization. It's better to have a wall clock time comparison on the same hardware (again with a few repeats) to understand real-world gains.”
> > >
> > > Reviewer Comment:
> > > “Report wall time in addition to FLOPS, to help understand e2e impact wrt hardware optimization.”
> >
> > **Response:**
> > We thank the reviewer for pointing this out. We have conducted a deep layer-level latency profiling on the same hardware (H100) and added the wall-clock breakdown to Appendix A.5.
> >
> > The execution of a pure baseline LLaDA-8B block takes roughly 1.453 ms. The total overhead introduced by our dynamic router is approximately 0.232 ms. Crucially, the pure neural network computation of the router consumes only $\sim$0.087 ms (a mere 6.0\% of the baseline block's latency). The remaining majority of the overhead ($\sim$0.145 ms) is entirely dominated by framework-level artifacts required to manage dynamic execution within PyTorch (e.g., memory gather/scatter and GPU-CPU synchronization).
> > This profiling highlights that the pure algorithmic complexity of DIALS is minimal. We have clarified in the text that with further optimizations of custom kernels for dynamic execution, the actual wall-clock latency is expected to converge closely toward our theoretical FLOPs reduction bounds.
> >
> > > Q6. Additional related work: ES-dLLM, AdaDiff, and DeeDiff
> > > Reviewer Comment:
> > > “In addition to that, with full transparency, I also used an AI agent to help check related works. It suggested that ES-dLLM and AdaDiff/DeeDiff could be significantly related work. I'm aware that some work could have appeared after this paper was written. Please consider adding them to your related work.”
> >
> > **Response:**
> > We thank the reviewer for pointing this out. We have carefully reviewed these suggestions. ES-dLLM proposes an inference acceleration strategy utilizing caching mechanisms, which is relevant to our broad motivation of efficient inference; accordingly, we have cited and discussed it in the related work section. Conversely, AdaDiff and DeeDiff operate on fundamentally different paradigms (they do not target Diffusion Language Models, nor do they focus on structural layer-skipping). Therefore, to maintain a focused discussion on DLM acceleration, we opted not to include them.

---

### Review · Reviewer_YfDG · 2026-05-21

**Summary Of Contributions:**

The paper proposes DIALS, a novel Dynamic Layer-Skipping framework for Diffusion Language Models. While dynamic layer skipping in autoregressive models operates at the token level, DIALS makes skip decisions on sequence of tokens. To achieve this, DIALS proposed a lightweight, two-layer MLP router between the transformer layers. These routers decide whether to skip the successive layer by calculating the average skip decision over the masked tokens and it crosses a specific threshold. Experiments conducted using the LLaDA-8B model demonstrate that DIALS achieves a better FLOPs-to-accuracy ratio compared to both static and random skipping baselines.

**Audience:**

Yes

**Audience Explanation:**

I believe his paper is highly relevant to the community as it addresses the high computational overhead of finetuning DLMs

**Broader Impact Concerns:**

I think there are no ethical concerns

**Claims And Evidence:**

Yes

**Claims Explanation:**

The proposed method shows a better FLOPs-to-accuracy, particularly on the PIQA and MMLU datasets. Furthermore, the paper provides  insights into layer importance through a thorough layer execution analysis

**Requested Changes:**

1. The threshold $\tau$ is set to 0.1 for all experiments. However the motivation for this choice is unclear. Please explain how varying this threshold affects the performance.
2. Equation 5 computes average skip proportion over all masked token scores to make a decision. Please clarify why averaging was selected over other intuitive strategies like majority voting.
3. The paper sets $\alpha_{\text{target}}$ to 0.5, but the paper do not provide any progression analysis of $\alpha$ during training. Did $\alpha$ converge to $\alpha_{\text{target}}$ at the end of training.

---

> ### Author Response · Authors · 2026-06-05
> **Author Response for Reviewer YfDG**
>
> Thanks for your insightful feedback and valuable review.
> > Q1. Motivation and sensitivity analysis of the routing threshold $\tau$
> > Reviewer Comment:
> > “The threshold $\tau$ is set to 0.1 for all experiments. However the motivation for this choice is unclear. Please explain how varying this threshold affects the performance.”
>
> **Response:**
> We thank the reviewer for pointing this out. The value $\tau=0.1$ was chosen empirically as it provided a stable baseline performance across various tasks during our initial experiments. To directly address your concern regarding sensitivity, we have conducted a new ablation study varying $\tau$ and added the results to Appendix A.3. As shown in the updated Appendix, varying $\tau$ does not yield a uniquely optimal configuration. Instead, the results consistently follow the standard efficiency-accuracy trade-off, exhibiting behavior fundamentally comparable to varying the scaling factor $\lambda$. This confirms that $\tau=0.1$ is not a strictly sensitive or lucky choice, but rather it simply serves as an alternative hyperparameter that influences where the model converges on the efficiency-accuracy trade-off curve.
>
> > Q2. Rationale for using average skip proportion instead of majority voting
> > Reviewer Comment:
> > “Equation 5 computes average skip proportion over all masked token scores to make a decision. Please clarify why averaging was selected over other intuitive strategies like majority voting.”
>
> **Response:**
> We thank the reviewer for pointing this out. We would like to clarify that our formulation---averaging the discrete token-level votes and comparing them against a continuous threshold $\tau$---is actually a direct generalization of majority voting.
>
> To briefly revisit the mathematical mechanism in Equations 5 and 6: the router outputs a discrete vote ($v_{l,0}^{(i)} \in \{0, 1\}$) for each token, and Equation 5 computes the average $\bar{v}_{l,0}$, which is exactly the percentage of tokens voting to skip the layer. Crucially, if we set the threshold $\tau = 0.5$, our decision mechanism becomes mathematically identical to strict majority voting (i.e., the layer is skipped if more than 50\% of the tokens vote to skip).
>
> We selected this generalized formulation simply to introduce $\tau$ as a configurable variable, ensuring the framework remains adaptable rather than restricted to a hard-coded 50\% rule.
>
> > Q3. Progression and convergence analysis of $\alpha$ during training
> > Reviewer Comment:
> > “The paper sets $\alpha_{\mathrm{target}}$ to 0.5, but the paper do not provide any progression analysis of $\alpha$ during training. Did $\alpha$ converge to $\alpha_{\mathrm{target}}$ at the end of training.”
>
> **Response:**
> We thank the reviewer for pointing this out. We have added a progression analysis to Appendix A.1 (Figure 7), presenting side-by-side plots of the layer execution ratio trajectories under different $\lambda$ values.
>
> As shown in the figures, the execution ratio smoothly converges to a stable value. It does not converge strictly to $\alpha_{\mathrm{target}}$ ($0.5$), because the final ratio represents a dynamic balance between preserving accuracy (task loss) and reducing FLOPs (penalty). The side-by-side comparison clearly illustrates that increasing $\lambda$ actively pulls this convergence point closer to the target (e.g., shifting from $\sim 0.96$ down to $\sim 0.90$).
>
> Thus, $\alpha_{\mathrm{target}}$ acts as a directional anchor rather than a rigid constraint, allowing the router to naturally find an optimal efficiency-accuracy balance without performance collapse.

---

### Decision · Action_Editor_wuct · 2026-07-09

**Recommendation:** Reject

**Additional Comments:**

I want to encourage the authors to continue working on this direction with a major revision in a resubmission, as the issues mentioned above are significant but repairable still. Specifically, I would like the authors to provide
1. More careful experimental results with error bars so we can compare methods more properly
2. More careful measurement of wall clock time

Also I would like the authors to tone down the claims that are not well supported by evidence, as in my opinion, it is not necessary for the method to outperform in every category in order to accept this paper. I would much rather have an accurate reporting of a promising method along with all of its flaws carefully analyzed.

**Audience:**

Yes

**Audience Explanation:**

As all the reviewers agree, the study of layer skipping techniques for DLMs are valuable. Even if the results are moderate or negative, they still may be of interest to many readers of TMLR.

The authors are very convincing with presenting the problem: DLMs cannot naively skip layers the same way that usual Transformers do, as the tokens are not being generated in sequence. New methods should be studied, and if the results are negative, the idea is still tested and now exchanged to be built on in the future.

**Claims And Evidence:**

No

**Claims Explanation:**

As reviewers qkhg and L8FN mentioned, there are several issues I believe that are not yet convincing in this paper
1. The method proposed does not seem to have strong generalization
2. Dynamic layer skipping does not seem to improve from statistic skipping
3. Wall clock time does not seem to improve despite the FLOPs savings

It's definitely too harsh to say the authors have no evidence towards these issues, but I agree with the reviewers that I do not find them sufficiently convincing at this point. I believe that a more modest reframing of the work towards a diagnostic paper, and only claim improvements when there are convincing evidence, can still make this a valuable paper.

**Resubmission Of Major Revision:**

The authors may consider submitting a major revision at a later time.